# Support Models for Addiction Related Treatment (SMART) for pregnant women: Study protocol of a cluster randomized trial of two treatment models for opioid use disorder in prenatal clinics

Ariadna Forray[1]*, Amanda Mele[1], Nancy Byatt[2,3,4], Amalia Londono Tobon[5], Kathryn Gilstad-Hayden[1], Karen Hunkle[1], Suyeon Hong[1], Heather Lipkind[6], David A. Fiellin[7,8,9], Katherine Callaghan[3], Kimberly A. Yonkers[2]

1 Department of Psychiatry, Yale School of Medicine, New Haven, Connecticut, United States of America,
2 Department of Psychiatry, University of Massachusetts School of Medicine, Worcester, Massachusetts, United States of America, 3 Department of Ob/Gyn, University of Massachusetts Medical School, Worcester, Massachusetts, United States of America, 4 Department of Population and Quantitative Health Sciences, University of Massachusetts Medical School, Worcester, Massachusetts, United States of America, 5 Department of Psychiatry and Human Behavior, Brown University, Providence, Rhode Island, United States of America, 6 Department of Obstetrics, Gynecology, and Reproductive Sciences, Yale School of Medicine, New Haven, Connecticut, United States of America, 7 Department of Internal Medicine, Yale School of Medicine, New Haven, Connecticut, United States of America, 8 Department of Emergency Medicine, Yale School of Medicine, New Haven, Connecticut, United States of America, 9 Yale School of Public Health, New Haven, Connecticut, United States of America

* ariadna.forray@yale.edu

**Funding:** This work was supported by Patient-Centered Outcomes Research Institution (PCORI)

## Abstract

### Introduction

The prevalence of opioid use disorder (OUD) in pregnancy increased nearly five-fold over the past decade. Despite this, obstetric providers are less likely to treat pregnant women with medication for OUD than non-obstetric providers (75% vs 91%). A major reason is many obstetricians feel unprepared to prescribe medication for opioid use disorder (MOUD). Education and support may increase prescribing and overall comfort in delivering care for pregnant women with OUD, but optimal models of education and support are yet to be determined.

### Methods and analysis

We describe the rationale and conduct of a matched-pair cluster randomized clinical trial to compare the effectiveness of two models of support for reproductive health clinicians to provide care for pregnant and postpartum women with OUD. The primary outcomes of this trial are patient treatment engagement and retention in OUD treatment. This study compares two support models: 1) a collaborative care approach, based upon the Massachusetts Office-Based-Opioid Treatment Model, that provides practice-level training and support to providers and patients through the use of care managers, versus 2) a telesupport approach

(grant number MAT-2018C2-12891) awarded to AF and KY. PCORI had no role in the design of this study and will not have any role during its execution, analyses, interpretation of the data, or decision to submit results. LT performed this work while employed by Yale University and Lifespan/Brown University; however, she is now employed by the Division of Intramural Research, National Institute of Minority Health and Health Disparities, National Institutes of Health. The contents and views in this manuscript are those of the authors and should not be construed to represent the views of the National Institutes of Health. The funders had and will not have a role in study design, data collection and analysis, decision to publish, or preparation of the manuscript. Funder Contact: Andrea Brandau Program Officer Patient-Centered Outcomes Research Institute (PCORI) 1828 L Street, NW 9th Floor, Washington, DC 20036 Office: 202.370.9394 abrandau@pcori.org https://www.pcori.org/people/andrea-brandau-mpp.

**Competing interests:** I have read the journals policy and the authors of this manuscript have the following competing interests: Dr. Nancy Byatt has received salary and/or funding support from Massachusetts Department of Mental Health via the Massachusetts Child Psychiatry Access Program for Moms (MCPAP for Moms). She is also the statewide Medical Director of MCPAP for Moms and the Executive Director of Lifeline4Moms. She has served on the Medscape Steering Committee on Clinical Advances in Postpartum Depression. She received honoraria from Medscape, Miller Medical Communications and Mathematica. She has served on the Perinatal Depression Advisory Board for the Janssen Disease Interception Accelerator Program, Advisory Boards for Sage Therapeutics. She has also served as a consultant Ovia Health, Sage Therapeutics or their agents, and has received speaking honoraria from Sage Therapeutics. Dr. Kimberly Yonkers has consulted to Athenen Pharmaceuticals. Dr. Leena Mittal consulted to Sage Therapeutics. This does not alter our adherence to PLOS ONE policies on sharing data and materials.

based on the Project Extension for Community Healthcare Outcomes, a remote education model that provides mentorship, guided practice, and participation in a learning community, via video conferencing.

## Discussion

This clustered randomized clinical trial aims to test the effectiveness of two approaches to support practitioners who care for pregnant women with an OUD. The results of this trial will help determine the best model to improve the capacity of obstetrical providers to deliver treatment for OUD in prenatal clinics.

## Trial registration

**Clinicaltrials.gov trial registration number**: NCT0424039.

## Introduction

Between 1999 and 2014 the point prevalence of an opioid use disorder (OUD) in pregnant patients increased from 1.5 to 6.5 per 1000 deliveries, which is in line with rates of opioid prescribing in the general population [1]. Thus, it is not surprising that between 2007 and 2016, pregnancy-associated mortality resulting from overdose more than doubled in the US, from 1.3 to 4.2 deaths per 100,000 live births. In many states, overdose is the leading cause of maternal morbidity and mortality [2]. Unfortunately, outpatient providers of medication treatment for OUD (MOUD) are less likely to treat pregnant patients compared to non-pregnant patients (75% vs 91%) [3].

Medication for OUD (MOUD) and behavioral therapy/psychosocial support are recommended in this population [4–10]. Medication treatment enhances adherence to prenatal care and reduces pregnancy and birth complications [11]. The use of MOUD in pregnancy increases the likelihood that a woman will continue treatment after delivery, a period of high risk among perinatal patients with OUD [12]. Unfortunately, it is often difficult for perinatal patients with OUD to access care because of challenges in treatment availability (wait lists, absence of specialized addiction programs, centers that will not enroll pregnant patients) [3,13,14]; accessibility (limited transportation, competing time demands given child care) [15]; affordability (lack of insurance or other financial resources) [16]; and acceptability (concerns over reports to child protective services, stigma/shame related to the illness, provider attitudes toward illness and MOUD) [13,17].

Given the current scope of the opioid crisis in the US, the need for treatment outpaces the capacity to provide it [18]. Obstetrical providers can enhance treatment capacity but report a variety of barriers including lack of expertise and education [19–22], limited physician time and resources [19,23,24], concern about MOUD misuse or diversion [18,25], lack of institutional support [18,23,26], cumbersome regulations [21,27], and insurance barriers (e.g. insufficient rates of reimbursement) [18,19,22,24].

The use of MOUD in perinatal patients is one component of treatment. Psychosocial support and sensitive, respectful approaches to care are also requisite and can enhance retention in treatment and thereby improve maternal and fetal well-being. However, to deliver this level of care, providers need education and training. Unfortunately, little information is available on the best systems of care to provide expertise and support to prenatal care providers and

their patients with OUD. To this end we designed a study to compare the effectiveness of two models of support for reproductive health clinicians who provide care for pregnant and postpartum patients with OUD: 1) collaborative care (CC) and 2) Project Extension for Community Healthcare Outcomes (Project ECHO). The CC approach is based upon the Massachusetts Office-Based-Opioid Treatment (OBOT) Model [28] that provides practice-level training and support to providers and patients through the use of care managers (CMs). Project ECHO is a telesupport remote education model that provides mentorship, guided practice, and participation in a learning community, via video conferencing [29]. Both models show feasibility and acceptability in primary care settings but have not been studied in obstetrical settings.

This paper aims to describe the Support Models for Addiction Related Treatment (SMART) trial, a matched pair cluster-randomized clinical trial protocol designed to compare two support models (CC vs. Project ECHO) that provide buprenorphine education and support for providers caring for pregnant patients with OUD.

## Methods and design

The institutional review board (IRB) at all centers participating in this study approved the following study, including Yale University IRB, Lowell General Hospital IRB, Hartford Healthcare IRB, and Beth Israel Deaconess Medical Center IRB. At Yale, the SMART Trial was approved on 1/20/2020 by the Human Research Protection Program Institutional Review Boards, FWA00002571, IRB Protocol ID: 2000027031, submission ID: 200002703. Written informed consent will be obtained from all participants. Study findings will be disseminated through peer-reviewed publications and presentations at scientific conferences.

### Study aims and overview

This study protocol was developed in response to a Patient-Centered Outcomes Research Institute (PCORI) funding initiative addressing the following question: "What is the comparative effectiveness of different strategies for providing support or coordination of services for components of medication-assisted treatment (MAT) (induction and/or psychosocial services) to providers who offer office-based opioid treatment (OBOT) to pregnant women, in terms of maternal and neonatal outcomes?" (https://www.pcori.org/funding-opportunities/announcement/medication-assisted-treatment-cycle-2-2018).

There are two primary aims in this study. First, to determine differences in engagement and retention in OUD treatment (MOUD and/or non-pharmacological care) between patient participants who receive care from a center that uses a CC vs. Project ECHO support model (Aim 1). The second primary aim, is to determine differences in patient activation according to the Patient Activation Measure (PAM). Activation has four stages; "believing the patient role is important, having the confidence and knowledge necessary to take action, actually taking action to maintain and improve one's health, and staying the course even under stress" [30].

The SMART trial includes twelve obstetric centers in Connecticut (n = 8) and Massachusetts (n = 4) that were randomized to Project ECHO or CC support models modified for perinatal women.

### Study conditions

**SMART ECHO.** Project ECHO (Extension for Community Healthcare Outcomes) was developed originally to build capacity to treat chronic, complex health conditions in rural and underserved communities that lack ready access to clinical specialists [31,32]. It uses a virtual

hub-and-spoke educational model that links primary care clinicians with specialists through a real-time learning model made possible by inexpensive videoconferencing technology [31,33]. Unlike traditional telemedicine, the ECHO model results in "force multiplication" [34], where a few specialists mentor many providers, who in turn provide enhanced care for large numbers of patients that would otherwise have limited access to specialty care. Project ECHO is now used to train providers to manage many other conditions, such as HIV [35,36], mental illness [37,38], chronic pain [39,40], diabetes [41], and OUD [33]. Data show it to be an effective and potentially cost-saving model [33]. The Project ECHO Integrated Addictions and Psychiatry (IAP) program trained the largest number of buprenorphine-waivered physicians in rural areas of New Mexico, which started at <20 and increased to 140 as of 2014 [42].

For SMART ECHO, we modified the IAP ECHO to allow expert obstetricians and psychiatrists who regularly treat perinatal women for OUD to train other obstetric providers about OUD screening, diagnosis, and treatment in pregnancy. We collaborated with the Weitzman Institute (https://www.weitzmaninstitute.org/project-echo) to develop SMART ECHO. The Weitzman Institute is a certified "replication center" for Project ECHO, which ensures that all elements of the ECHO sessions are in accordance with the evidence-based approach developed and promoted by the University of New Mexico. Consistent with the traditional ECHO model, we have one hub for all study practices randomized to this condition. A typical SMART ECHO session consists of 1) introductions of participants; 2) a brief didactic session, usually a 30-minute presentation on substance use or mental health; and 3) discussion of case presentations submitted by participants in advance for one hour total. Sessions are twice a month in the first six months and monthly thereafter. Examples of topics covered in these sessions include, management of OUD in pregnancy, pain management and anesthesia at delivery, postpartum management of OUD, neonatal opioid withdrawal syndrome, harm reduction strategies, and psychiatric comorbidities.

**Collaborative Care for Opioids in Pregnancy (CC-OP).** CC was originally developed to enhance the capacity for treatment of depression in primary care settings [43]. As articulated by the University of Washington, it includes several components [1,31]: systematic screening for medical and behavioral health needs [3,33] and a team-based approach that includes a care manager (CM), psychiatrist and primary care physician [44,45]. The CC model has since been refined [46,47] and tested for treatment of other disorders; it was deployed in additional venues including obstetrical-gynecological settings [48,49]. CC was used in the treatment of OUD in open trials of non-pregnant patients [50,51] and pregnant women [52]. It increased treatment initiation, engagement, and use of psychotherapy among non-pregnant patients [53].

For our CC-OP, we generally follow the Collaborative Opioid Prescribing Model (Massachusetts OBOT Model). However, we are retaining several features of the original CC model that were not in the Massachusetts OBOT Model including the use of a registry. The registry is a secure web-based patient tracking system that is updated twice a month by the CM. Also, we will not limit inclusion to women only selecting buprenorphine treatment. CC-OP includes the following components: universal screening of all pregnant women, a team-based approach where a CM and the obstetrical provider discuss each patient at least once every two-weeks, treating to wellness with regular monitoring via a patient registry, and recovery coaching. Universal screening is done via a validated tool (e.g. the NIDA quick screen, the 4P's Plus, the SURP-P, etc.), selected by the individual practices and integrated to the individual practice workflow. The CM responsibilities include: 1) screening patients for an OUD; 2) entering CC-OP participants into the patient registry; 3) providing education to participants about OUD; 4) assistance in initiation procedures for patients who would like to receive buprenorphine; 5) informing the obstetrical provider of a positive urine drug screen and need to consider an increase in buprenorphine; if the patient is on methadone they coordinate with the

outside treatment program to ensure adequate intervention/support; 6) if the patient is not receiving MOUD, discuss its role and/or psychological treatment; and 7) providing recovery coaching.

## Practice recruitment

Obstetrical clinics of any type, private practices, hospital-based clinics, or Federally Qualified Health Centers, were approached by the study principal investigators. Sites were recruited based on the number of cases of OUD in the geographical area and their ability to be randomized to one or the other condition. The parts of Connecticut and Massachusetts that were of interest were those that had high need. Sites and physicians voluntarily participated. There were no practice eligibility criteria or ineligibility criteria other than their ability to be randomized and willingness to participate.

## Practices and allocation procedures

The unit of randomization was the obstetrical practice. Twelve obstetrical centers were matched into six pairs according to state (4 in Massachusetts and 8 in Connecticut), size, and academic (6) vs non-academic (6), private practices. Most practices were located in smaller urban areas. To randomly assign centers to a model of care, the statistician, masked to the identities of the sites, assigned numbers to each center using a random number generator. Within each matched pair, the clinic assigned the lower randomly generated number was allocated to CC-OP; the remaining clinic was assigned to SMART ECHO for a total of 6 clinics per model of care. The collaborating physicians identify a part-time mid-level clinician, such as a nurse, social worker or research assistant, who is trained to be the CM for CC-OP or data support person for SMART ECHO.

## Patient participant eligibility/ineligibility criteria

Patient participants follow their site randomization. To be eligible, patient-participants must be at least 18 years of age, speak/read English, have a diagnosis of DSM-5 OUD and be less than 34 weeks pregnant at the time of enrollment. Participants are not required to receive or be receiving MOUD at study entry but may be started on MOUD during the field study if they so choose.

## Outcomes

**Primary outcomes.** The first primary outcome is percent engaged which is operationalized as > 2 visits for opioid use disorder treatment in 30 days [53] (>2 visits within 30 days of baseline visit; < = 2 visits since baseline or did not consent to be in the study).

The second component of Aim 1 is retention. We define unsuccessful retention according to a modification of Wilder, et al. (2015) [12], as women who are enrolled in pharmacological or behavioral treatment for opioid use disorder who stopped treatment with no plan for ongoing therapy or medication for > one month (e.g. discharge, relapse and left treatment, lost to follow up with no discharge plan) during pregnancy, and the 3 months post-delivery. We hypothesize that the benefit from a CM and proactive monitoring approach will lead to greater treatment engagement and retention among patient participants.

For Aim 2, our primary outcome is the 13-item Patient Activation Measure (PAM) [30]. The PAM is a patient-centered questionnaire that measures health care knowledge, beliefs, skills, and confidence in managing illnesses, and uses a 4-point Likert scale with higher scores

showing more favorable health outcomes [30]. Further details on primary outcomes are listed on Table 1.

**Secondary and exploratory measures and outcomes.** We will collect secondary and exploratory process and outcomes measures that are outlined in Table 2 and Fig 1. Secondary and exploratory analyses for Aim 1 will compare eligible women in the two conditions (gravidas with an OUD, not in treatment when presenting for prenatal care) on the following outcomes: 1) #/% offered MOUD including buprenorphine, specifically; 2) #/% initiated onto MOUD; 3) for all participants, #/% retained on MOUD and 4) rates of abstinence from illicit opioids or misuse of prescription opioids [54]. Additionally, for all pregnant women treated with MOUD: 1) #/% continuing MOUD at 3 months postpartum; 2) #/% engaging in an opioid treatment program at 3 months post-delivery; 3) #/% with concurrent substance use according to the Timeline Followback (TLFB) [54] and urine tests; 4) fetal and neonatal outcomes (low birth weight, resuscitation at delivery, fetal demise, preterm birth, duration of hospitalization); 5) racial and ethnic differences; and 6) differences among participants who use illicit street opioids (e.g. heroin) vs. those that use prescription opioids.

Secondary and exploratory outcomes for Aim 2 include participant reported differences in: 1) the Shared-Decision Making Questionnaire-9 (SDM-Q-9) [55]; 2) Perceived efficacy in Patient-Physician Interactions Questionnaire (PEPPI) [56]; 3) Kim Alliance Scale-Communication subscale (KAS-CM) [57]; 4) PROMIS Emotional Short Form and Satisfaction with Roles and Activities [58]; and 5) Edinburgh Postnatal Depression Scale (EPDS) [59,60]; 6) Stigma-Related Rejection Scale (SRS) [61]. We will also explore racial and ethnic differences in patient measures and possible differences in outcomes among those who use illicit street opioids vs misuse of prescription opioids.

Providers complete the following secondary outcomes: the Autonomy (5 items: e.g. freedom in practice activities), Patient Care Issues (4 items: e.g. perception of needs of patients), and Relationship with Patients (4 items: e.g., satisfaction with patient relationship) subscales from the Physicians Worklife Survey (PWS) [62] and the Treatment Optimism Subscale of the Substance Abuse Attitude Survey (SAAS (4 items)) [63] that measures attitudes toward treatment of individuals with substance use.

## Fidelity procedures

**CC-OP.** To ensure that CC-OP is being implemented with fidelity we will:

1. Check screening rates for an OUD by conducting medical chart reviews of at least 50 consecutive charts (selected to start at a random date) at each CC-OP site at baseline, 6 months, 12 months, 18 months, 24 months, and 36 months. Screening is considered completed if patients were asked about substance use using a validated scale and documented in the medical record. The percentage of women being screened for OUD at their first prenatal visit is calculated.

**Table 1. Primary outcome measures.**

| Outcome | Definition | Assessment Tool | Assessment Timepoint(s) |
|---|---|---|---|
| Treatment engagement | > 2 visits for opioid use disorder treatment within 30 days of baseline | Treatment utilization form, medical records, and study database | 30 days after baseline |
| Treatment retention | No stoppage of OUD treatment (with no plan for ongoing therapy or medication) for > one month | Treatment utilization form, medical records, and study database | Delivery and 3-months postpartum |
| Patient Activation | Increase in at least one level on the Patient Activation Measure (PAM) from baseline to 34 weeks and 3-months postpartum. | Patient Activation Measure (PAM) | Baseline, 26 weeks, 34 weeks and 3-months postpartum |

**Table 2. Secondary and exploratory outcomes.**

| Outcome | Definition | Assessment Tool | Assessment Timepoints | Type of Outcome |
|---|---|---|---|---|
| | | Process Measures | | |
| Initiated onto medication for opioid use disorder (MOUD) | > 2 visits for MOUD treatment within 30 days of baseline | Treatment utilization form (TUF), medical records, and study database | 30 days after baseline | Secondary |
| Retained on MOUD | No stoppage of MOUD treatment (with no plan for ongoing therapy or medication) for > one month | TUF, medical records, and study database | Delivery and 3-months postpartum | Secondary |
| Offered MOUD | Healthcare provider discussed MOUD with patient | Patient self-report at baseline and monthly research surveys via TUF | 30 days after baseline | Exploratory |
| Postpartum Engagement in treatment | Engaged in an opioid treatment program at 3-months postpartum | TUF, medical records, and study database | Delivery and 3-months postpartum | Exploratory |
| Abstinent from illicit opioids or misuse or prescription opioids | No self-reported use and negative drug screen | a. Timeline Followback (TLFB) b. Urine drug screens | a. monthly assessments b. each prenatal and post-delivery office visit | Exploratory |
| Concurrent substance use | Self-reported use and/or positive urine drug screen | a. TLFB b. Urine drug screens | a. monthly assessments b. each prenatal and post-delivery office visit | Exploratory |
| | | Patient Measures | | |
| Stigma | Scale score | Stigma-Related Rejection Scale (SRS) | Week 26 of pregnancy and 3-months postpartum | Secondary |
| Shared Decision Making | Scale score | SDM-Q-9 | Baseline and week 26 and 36 of pregnancy | Exploratory |
| Patient-Physician Interaction | Scale score | PEPPI-5 | Baseline, week 26 and 36 of pregnancy, and 3-months postpartum | Exploratory |
| Clinician/Patient therapeutic relationship | Scale score | Kim Alliance Communication (KAC) | Baseline, week 26 and 36 of pregnancy, and 3-months postpartum | Exploratory |
| Satisfaction with roles and activities | Scale score | PROMIS Emotional Short Form 4a | Baseline, week 26 and 36 of pregnancy, and 3-months postpartum | Exploratory |
| Depression | Scale score | EPDS | Baseline, week 26 and 36 of pregnancy, and 3-months postpartum | Exploratory |
| | | Practitioner Measures | | |
| Physician Work Satisfaction | Scale score | Physician Work-Life Survey | Before first participant enrollment and after last participant enrollment | Secondary |
| Attitude Toward Treatment of Individuals with Substance Misuse | Scale score | Substance Abuse Attitude Survey (SAAS) | Before first participant enrollment and after last participant enrollment | Secondary |
| | | Birth Outcomes | | |
| Birth weight | Weight in grams at birth | Medical records | Birth | Secondary |
| Low birth weight | <2500 grams | Medical records | Birth | Exploratory |
| Resuscitation at delivery | Any respiratory assistance at birth: suctioning, positive pressure ventilation via bag/mask, endotracheal intubation, chest compression, epinephrine/volume administration | Medical records | Birth | Exploratory |
| Fetal demise | Intrauterine fetal demise after 20 weeks' gestation and/or 350 grams birthweight | Medical records | Birth | Exploratory |
| Preterm birth | Born before 37 weeks' gestation | Medical records | Birth | Exploratory |

(*Continued*)

**Table 2.** (Continued)

| Outcome | Definition | Assessment Tool | Assessment Timepoints | Type of Outcome |
|---|---|---|---|---|
| Duration of hospitalization | Discharge date- admittance date | Medical records | Hospital discharge | Exploratory |

Note: SDM-Q-9 = 9-item Shared Decision Making Questionnaire; PEPPI-5 = 5-item Perceived Efficacy in Patient-Physician Interactions; PROMIS = Patient-Reported Outcomes Measurement Information System; EPDS = Edinburgh Postnatal Depression Scale.

2. Measure CM and/or physician contacts the patient (either in-person, by phone or via tele-health) and discuss each their progress with OUD treatment at least biweekly. This is tracked in the participant registry, which is audited quarterly by the coordinating centers and is a measure of "team-based care."

3. Follow the "treat to wellness" model. If a participant's urine drug screen (UDS) is negative for a prescribed MOUD, or if the UDS is positive for illicit or non-prescribed opioids, action will be taken by the providers (e.g., increase buprenorphine dose, initiate or extend recovery coaching or refer to additional substance use treatment). Actions are noted in the registry and audited quarterly. We give CC-OP sites a "report card" every 6 months that provides feedback on adherence to the fidelity metrics.

**SMART ECHO.** To ensure fidelity to the Project ECHO model we will follow the Weitzman Institute's replication model [64]:

1. Use a web-based database to monitor outcomes. Physician participant and session details will be tracked in the UNM-required iECHO platform. Physician participants will complete an enrollment form on Survey Monkey before the first SMART ECHO session to capture demographic information, session topic ideas, pre-intervention self-reported measures of attitudes, self-efficacy, behavior related to screening, prescribing, and inter-disciplinary care. A program specialist will maintain an attendance tracker for all sessions, which will be completed after each session. Physician participants will complete a CME survey at the end of each session, as well as mid-series and end-of-series surveys to capture: overall satisfaction with SMART ECHO, session topic ideas, self-reported measures of attitudes, self-efficacy.

2. Follow a case-based learning format. The majority of each SMART ECHO session is dedicated to at least 30 minutes of case discussion. Participants are provided a case feedback form inquiring about their perceptions of the quality of the recommendations they received and the likelihood that they will use the recommendations provided. The content is monitored according to: # of unique cases presented, # of cases presented, and # (%) of participants who submitted cases.

3. Leverage technology and utilize Zoom videoconferencing and leverage its advanced features, including polling and chat, to promote engagement during SMART Project ECHO sessions.

4. SMART ECHO team uses the Project ECHO Quality Checklist, which assesses the quality of the following domains: technology, remote faculty technology, session logistics, faculty, participants, and HIPAA Compliance. This checklist is completed during each ECHO session and allows the measurement of the % of ECHO sessions that satisfied all of the quality requirements of a Project ECHO session.

| | Study Screening | Visit 1[1] Baseline | Visit 2[1] | Visit 3[1] | Visit 4[1] | Visit 5[1] 26-week Visit | Visit 6[1] | Visit 7[1] | Visit 8[1] 3-week Visit | Visit 9[1] | Visit 10[1] Birth | Visit 11[1] 3-mo PP[7] visit |
|---|---|---|---|---|---|---|---|---|---|---|---|---|
| **Timepoint** | | | | | | +/- 2 weeks | | | +/- 2 weeks | | | +/- 2 weeks |
| Informed Consent | X | | | | | | | | | | | |
| Validated substance use screening tool[2] | X | | | | | | | | | | | |
| Demographics | X | | | | | | | | | | | |
| Concomitant Medication | | X | X | X | X | X | X | X | X | X | | |
| Medical History | | X | | | | | | | | X | | |
| Pregnancy Questionnaire | X | | | | | | | | | | | |
| DSM-5 OUD Checklist[3] | X | | | | | | | | | | | |
| Urine Drug Screening | | X | X | X | X | X | X | X | X | X | | |
| Assigned OUD Treatment[2] | | X | X | X | X | X | X | X | X | X | | |
| Timeline Followback[4] | | X | X | X | X | X | X | X | X | X | | |
| Post-Traumatic Stress Disorder Checklist for DSM-5 (PCL-5) | | X | | | | | | | | | | |
| Fagerstrom Test for Nicotine Dependence (FTND) | | X | | | | X | | | X | | | X |
| HIV Risk -Taking Behavior Scale (HRBS) | | X | | | | | | | | | | |
| Patient Activation Measure (PAM) | | X | | | | X | | | X | | | X |
| Shared-Decision Making Questionnaire-9 (SDM-Q-9) | | X | | | | X | | | X | | | X |
| Perceived efficacy in Patient-Physician Interactions Questionnaire (PEPPI) | | X | | | | X | | | X | | | X |
| Kim Alliance Scale-Communication subscale (KAS-CM) | | X | | | | X | | | X | | | X |
| PROMIS Emotional Short Form 4a | | X | | | | X | | | X | | | X |
| PROMIS Satisfactions with Roles and Activities Short form 8a | | X | | | | X | | | X | | | X |
| Edinburgh Postnatal Depression Scale (EPDS) | | X | | | | X | | | X | | | X |
| Adverse Childhood Experience Questionnaire | | X | | | | | | | | | | |
| Stigma Related Rejection Scale | | | | | | X | | | | | | X |
| Treatment Utilization Form | | X | X | X | X | X | X | X | X | X | | |
| Birth Outcomes[5] | | | | | | | | | | | X | |
| **Substance Abuse Attitude Survey (Treatment Optimism)[6]** | | X[5] | | | | | | | | | | |
| **Physicians Worklife Survey (PWS)[6]** | | X[5] | | | | | | | | | | |
| Qualitative Interviews (both providers and patients) | | | | | | X | | | | | | X |

**Fig 1. SPIRIT schedule of enrollment, interventions and assessments.** 1. Participant will enter the study at different point of pregnancy. Therefore, it is likely that participants will complete varying numbers of visits between screening and the 26-week visit, and the 36-week visit and birth. This is expected and is not considered a protocol deviation. Visit 1 will follow immediately after the consenting process. 2. All participating centers will administer a validated substance use screening tool, such as the NIDA Quick Screen, 4Ps Plus or equivalent validated instrument as standard of care. The tool utilized is determined by the clinical site. 3. The DSM-5 OUD Checklist will be administered to any patients who have a positive substance use screen as part of the standard of care. 4. The Timeline Followback (TLFB) will be conducted monthly during the length of the study. The number of TLFB assessments will vary depending on when the patient enters the study and when they give birth. 5. Birth outcomes include: Low birth weight,

resuscitation at delivery, fetal demise, preterm birth, duration of hospitalization. **6. Provider measures**: The Substance Abuse Attitude Scale, the Physicians Worklife Survey and the Qualitative Interviews will be completed by providers and done prior to the first participant being enrolled at the site and after the last participant is enrolled at the site. 7. **3**-mo PP: 3-months postpartum.

### Study procedures

Patients who screen positive for substance use and have OUD according to DSM-5 criteria will be informed about the study and if interested, site staff will obtain consent for further eligibility determination. After providing informed consent, patient-participants will complete a screening assessment that includes demographic information, pregnancy dates, a checklist that confirms an OUD diagnosis, and verification of other inclusion criteria and lack of exclusion criteria. If consent is not provided, only basic demographic information, number of weeks pregnant and the reason for disinterest, if known, is collected. Data will be collected on a secure tablet computer or via a secure email link using REDCap, a web application for building and managing online surveys and databases. A study team member will review and verify screening results to confirm eligibility. Participants will be compensated with an Amazon gift card for the time they spent during the screening process. Enrollees will complete additional baseline questionnaires and submit a urine sample for a UDS during Visit 1 and are compensated with an additional gift card for their time.

After completion of the baseline portion, patient-participants will be contacted monthly by a research assistant blinded to the patient's clinic or study arm. They will collect the Timeline Follow-Back (TLFB) [65] and the Treatment Utilization Form that includes questions about the quantity and frequency of all substances used and any treatment received outside of the study. Patient participants will complete additional self-report questionnaires at baseline, 26 weeks and 36 weeks (+/- 4 weeks), and at 3-months post-partum. Involvement of child protective services is assessed in the 3-month postpartum questionnaire. Maternal/infant medical record data will be extracted by the onsite staff for maternal, fetal, and neonatal outcomes (Fig 2).

### Analytic plan, power, and sample size estimate

Descriptive statistics will summarize data on all randomized subjects by treatment group, and also by site. Data analyses will employ the intent-to-treat principle and include follow-up on all patient-participants regardless of their retention to treatment.

To test for differences between care models for the primary outcomes, we will use permutation tests [66,67] that account for the cluster-randomized nature of the trial. We will test for differences in proportions (for binary outcomes) and in means (for quantitative outcomes) between the two interventions. The permutation approach will also be used to construct 95% confidence intervals for those differences.

To evaluate the different effects associated with implementation of each model, we will also report estimates of proportion and mean differences within each matched pair together with 95% confidence intervals. Due to feasibility constraints (only 12 groups in 6 matched pairs), we will not be able to perform a rigorous statistical evaluation of the effects of site-level factors. However, we will present descriptive statistics, by site, that may be useful in guiding future use of the proposed models.

In addition, we will perform exploratory mixed model analyses to compare each model's outcomes due to individual-level covariates including age, race/ethnicity, parity, illicit substances vs non-medical use of prescription opioids, and concurrent other substance use. Mixed effects models will account for potential positive correlations among observations

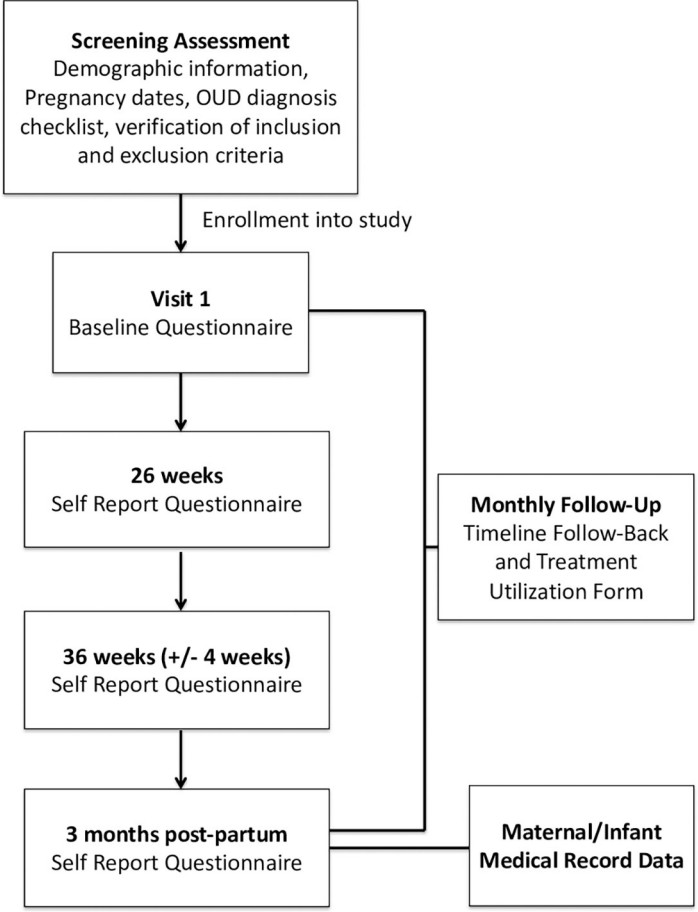

**Fig 2. Participant flow through study procedures.** Note: If a patient-participant is enrolled after 26 weeks' gestation and before 36 weeks' gestation, the items from the 26-week visit are added to the baseline assessment.

within sites and within matched pairs and will evaluate fixed effects for the interventions and potential covariates at the individual level. Generalized linear mixed models will be used for categorical outcomes and linear mixed models for continuous outcomes.

The proposed sites are primarily in high-need, urban settings. Combined, the sites in our study have an average of 36,995 deliveries per year. Based on the rates of OUD in pregnant women in MA (13.1/1000) and CT (10/1000) this would mean an average of 430 pregnant women per year with an OUD would be served by these locations for a total of 1,075 eligible women (or 90 per site) over the 2.5 years of study recruitment.

We used the approach of Hayes and Bennett (1999) [68] for matched-pair cluster randomized trials to conduct power analyses for our three primary dichotomous outcomes. For the treatment retention outcome, we hypothesized that the proportion retained in CC-OP would be greater compared to ECHO. A sample of six cluster pairs (12 clusters) with 40 participants per cluster achieves 80% power to detect a difference of 0.20 between the CC proportion engaged in treatment (0.94) [52] and the ECHO proportion (0.74). This is based on a two-sided paired test of the proportion difference with a significance level of 0.05 and an estimated within-pair coefficient of variation (CV) of 0.10 between clusters. We did not have empirical data for an estimate of the CV, so we followed the approach of Hayes and Bennett (1999) [68].

We generated 10,000 data sets with 6 clusters each and calculated the CV based on formula(9) in the manuscript. This follows the conservative approach suggested on p.322 of the paper (i.e. to use the coefficient of variation between clusters within each group as an upper limit for the coefficient of variation in matched pairs designs). After examining the distribution of the resulting coefficient of variation (CV) we selected 0.10 as the value for our subsequent power calculations since more than 90% of the data sets had values under that threshold.

For the treatment engagement outcome, we hypothesized that the proportion engaged in CC-OP would be greater compared to ECHO. Assuming 90 eligible participants per cluster with six clusters per group and a 43% engagement rate in the CC-OP group [53], we are powered at 80% to detect a 13% difference between CC-OP and SMART ECHO in engagement rates, setting significance level at 0.05 and CV at 0.10.

In an ongoing study of mothers with depression randomized to cognitive behavioral therapy with or without a community ambassador, 37% of 103 participants increased their patient activation by at least one PAM level (M. Smith, 2019, personal communication). Using this proportion for the CC-OP group and hypothesizing that participants in the CC-OP are more likely than ECHO participants to see an increase in patient activation, 40 participants per site will provide 80% power to detect a clinically meaningful 15% difference in the proportion of women who increase at least one PAM level between CC-OP and SMART ECHO. This estimate assumes a two-sided test, with a significance level of 0.05 and an estimated within-pair coefficient of variation (CV) of 0.10 between clusters.

### Provider and qualitative interviews

We will use qualitative methods to understand the experiences of obstetric care providers. We will invite at least one provider from each practice (minimum of 12) to participate in qualitative interviews to assess their perceptions and the impact of the care models to which they were assigned. A subset of patient participants (1–2 per site) will be given the opportunity to participate in qualitative interviews to collect data on patient satisfaction with treatment of OUD. Interviews will be transcribed by a member of the research team and de-identified. Further details of the qualitative work will be published in a forthcoming separate manuscript.

## Discussion

Improving the capacity of obstetric caregivers to provide treatment for pregnant patients with an OUD is critical. While addiction treatment admissions among pregnant patients with OUD increased over the past few years [69–71], the use of MOUD remains low [70,71], and the consequences are significant with opioid overdose becoming a leading cause of maternal mortality [2]. Barriers to treatment access are well documented [3,15], including a limited number of providers who offer and feel comfortable providing treatment for OUD in pregnancy. Many obstetricians who would like to offer comprehensive care to pregnant patients with OUD have not been given the knowledge base and support required to treat these patients. Addressing these barriers by providing education, training, and support can improve care for pregnant patients with OUD [15,52,72,73]. The Project SMART trial aims to address these limitations by testing the effectiveness of two approaches to support practitioners who care for women with OUD in obstetric settings.

Despite the potential benefits of a program that provides integrated prenatal and addiction care, there are a limited number of studies that compare the effectiveness of models for supporting obstetric settings. Only a few studies assessed possible differences in maternal and fetal well-being for gravidas treated in programs that provided combined vs. separate prenatal care and addiction treatment [72]. Some cohort studies provided information on fetal outcomes in

centers that provide combined care [73,74]. The results of this study can inform how we can better address OUD in obstetric settings and help address maternal mortality due to OUD and the issue of lack of care addressing social determinants of health. Equally important is the lack of knowledge regarding women's experiences when they receive treatment. This trial can provide information on patient's engagement in care, knowledge of care, confidence in managing pregnancy and their addiction, alliance with their providers, and emotional well-being, and fill in these important gaps.

The comparator conditions, SMART ECHO and CC-OP have different strengths and weaknesses. SMART ECHO is scalable, and with the development of a "learning community" through the group case presentation process, it is a very powerful way in which to share knowledge and enhance therapeutic confidence among providers. The benefits of the "learning community" can go beyond clinical issues of medication initiation and titration to include ways to improve referral systems. Depending upon the commitment of the practice and their willingness to share resources, multiple individuals—such as care providers and support staff—can be trained through the SMART ECHO model. However, members of the practice who participate in SMART ECHO, must make time and be available to attend the SMART ECHO sessions and to share case information with their colleagues. Availability can be a challenge in busy obstetric settings not only because of issues of patient flow but because of obstetrical emergencies. Additionally, physicians or advanced practice nurses must fit their treatment of OUD in practices that are already stressed for time.

Compared to SMART ECHO, CC-OP is advantageous in that its use of a CM may help to save physicians' time. The CM is trained in supporting behavioral techniques such as recovery coaching. The limitation of CC-OP is that it is potentially more labor-intensive—the CM must be willing to follow a group of patients rather closely and this can take time and effort. CC-OP can also be more costly than SMART ECHO; for the model to work, practices must be willing to dedicate financial resources to the CM. However, Medicare has made the costs of a CM billable and Massachusetts Medicaid has started to allow providers to bill for recovery coaching. Unfortunately, Medicaid reimbursement for recovery coaching is not universal and currently is not in place in Connecticut; a study such as this one could inform policymakers in funding these services.

The results of this Project SMART Trial may have policy and public health implications. Both the SMART ECHO and CC-OP interventions have the potential to be feasible, sustainable, and transportable to practice settings besides obstetric care. Measures of physicians' attitudes, perception of support, and perception of the relationship with patients will also provide data to understand clinicians' needs and potential targets for interventions. Lastly, the outcomes of this study may be helpful to payers and providers to understand how to implement adequate and manageable support systems. Patients, clinicians, payers, and policymakers will benefit from a comparison of these two models as we seek to understand if retention and engagement of pregnant women with opioid use disorder are impacted by the education and support their providers receive and the type of care patients receive during and after pregnancy.

## Supporting information

**S1 Checklist. Standard Protocol Items: Recommendations for Interventional Trials (SPIRIT) checklist.**
(DOC)

**S1 File. Approved Yale University IRB protocol.**
(PDF)

**S2 File. Approved Yale University IRB consent form.**
(PDF)

## Acknowledgments

The authors would like to thank Dr. Leena Mittal, Linda Brenckle, Dr. Joy Kaufman, Dr. Ralitza Gueorguieva, Dr. Kelsey White and Carolyn Friedhoff for their contributions to the conception or design of the work and edits to the final version of the manuscript.

## Author Contributions

**Conceptualization:** Ariadna Forray, Kimberly A. Yonkers.

**Data curation:** Ariadna Forray, Kathryn Gilstad-Hayden, Kimberly A. Yonkers.

**Formal analysis:** Ariadna Forray, Kathryn Gilstad-Hayden, Kimberly A. Yonkers.

**Funding acquisition:** Ariadna Forray, Kimberly A. Yonkers.

**Investigation:** Ariadna Forray, Amalia Londono Tobon, Kimberly A. Yonkers.

**Methodology:** Ariadna Forray, Kathryn Gilstad-Hayden, Kimberly A. Yonkers.

**Project administration:** Ariadna Forray, Karen Hunkle, Heather Lipkind, Katherine Callaghan, Kimberly A. Yonkers.

**Supervision:** Ariadna Forray, Karen Hunkle, Kimberly A. Yonkers.

**Writing – original draft:** Ariadna Forray, Amanda Mele, Amalia Londono Tobon, Karen Hunkle, Heather Lipkind, David A. Fiellin, Katherine Callaghan, Kimberly A. Yonkers.

**Writing – review & editing:** Ariadna Forray, Amanda Mele, Nancy Byatt, Amalia Londono Tobon, Kathryn Gilstad-Hayden, Karen Hunkle, Suyeon Hong, Heather Lipkind, David A. Fiellin, Katherine Callaghan, Kimberly A. Yonkers.

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
