## [Decision Letter · Decision Letter 0]

24 Aug 2021

PONE-D-21-07384

Support Models for Addiction Related Treatment (SMART) for Pregnant Women: Study Protocol of a Cluster Randomized Trial of Two Treatment Models for Opioid Use Disorder in Prenatal Clinics

PLOS ONE

Dear Dr. Forray,

Thank you for submitting your manuscript to PLOS ONE. After careful consideration, we feel that it has merit but does not fully meet PLOS ONE’s publication criteria as it currently stands. Therefore, we invite you to submit a revised version of the manuscript that addresses the points raised during the review process.

We look forward to receiving your revised manuscript.

Kind regards,

Yukiko Washio, Ph.D.

Academic Editor

PLOS ONE

Journal Requirements:

3. Please include your tables as part of your main manuscript and remove the individual files. Please note that supplementary tables (should remain/ be uploaded) as separate "supporting information" files

"I have read the journals policy and the authors of this manuscript have the following competing interests: Dr. Nancy Byatt has received salary and/or funding support from Massachusetts Department of Mental Health via the Massachusetts Child Psychiatry Access Program for Moms (MCPAP for Moms). She is also the statewide Medical Director of MCPAP for Moms and the Executive Director of Lifeline4Moms. She has served on the Medscape Steering Committee on Clinical Advances in Postpartum Depression. She received honoraria from Medscape, Miller Medical Communications and Mathematica. She has served on the Perinatal Depression Advisory Board for the Janssen Disease Interception Accelerator Program, Advisory Boards for Sage Therapeutics. She has also served as a consultant Ovia Health, Sage Therapeutics or their agents, and has received speaking honoraria from Sage Therapeutics. Dr. Kimberly Yonkers has consulted to Athenen Pharmaceuticals. Dr. Leena Mittal consulted to Sage Therapeutics."

5. Please note that in order to use the direct billing option the corresponding author must be affiliated with the chosen institute. Please either amend your manuscript to change the affiliation or corresponding author, or email us at plosone@plos.org with a request to remove this option.

6. Please amend your list of authors on the manuscript to ensure that each author is linked to an affiliation. Authors’ affiliations should reflect the institution where the work was done (if authors moved subsequently, you can also list the new affiliation stating “current affiliation:….” as necessary).

Reviewers' comments:

Reviewer's Responses to Questions

**Comments to the Author**

1. Does the manuscript provide a valid rationale for the proposed study, with clearly identified and justified research questions?

Reviewer #1: Yes

Reviewer #2: Yes

2. Is the protocol technically sound and planned in a manner that will lead to a meaningful outcome and allow testing the stated hypotheses?

Reviewer #1: Yes

Reviewer #2: Partly

3. Is the methodology feasible and described in sufficient detail to allow the work to be replicable?

Reviewer #1: Yes

Reviewer #2: Yes

4. Have the authors described where all data underlying the findings will be made available when the study is complete?

Reviewer #1: Yes

Reviewer #2: No

5. Is the manuscript presented in an intelligible fashion and written in standard English?

Reviewer #1: Yes

Reviewer #2: Yes

6. Review Comments to the Author

You may also provide optional suggestions and comments to authors that they might find helpful in planning their study.

Reviewer #1: This study presents a protocol for a trial aimed to investigate best practices for supporting OB care teams to integrate OUD treatment into OB care for pregnant women. I am happy to hear this trial is funded and going to take place, as we are desperately in need of evidence-based approaches at the ‘practice’ level to support providers in addressing the overdose crisis, especially for a priority population like pregnant and parenting people. Below I outline some queries and suggestions to strengthen the manuscript.

TEXT

- For the SMART ECHO program, is there a set curriculum that will be carried out across instutitons? Will it be one ‘hub’ for all participating study practices? Or will there be mlultiple?

- For the CC-OP, the registry is broight up in the methods multiple times, coming off as critical to proper data collection as well as effectivesnes of the intervention. Later on in the manuscript it starts to become more clear what this registry is – but would be helpful earlier on. For example, who is in charge of keeping this registry?

- Will there be one case manager per site? Full time positon? What qualifications will the CM have?

- Under primary outcomes, it says ‘treatmnet for opioid misuse’ – to be consistent with methods, I believe this should be OUD

- Will the treatment retention outcome be censored at 3 months? How will you handle that the ‘end of the study’ is 3 months but people may start treatment at different times

- Also, it is imperative that a justification for why this study will stop measuring outcomes at 3 months postpartum, rather than 12 months, is recommended – given that 7-12 months is the highest risk time for postpartum women with OUD for overdose, as an example.

- How will you measure if a patient was ‘offered’ MOUD? By provider doucmentaiton – and then chart abstraction? If yes, how will you mitigate information bias?

- Will substance use recurrence be measured only by urine drug test, or also by self-report? Post COVID, we are doing a lot more telehealth, so how will you take that into consideration for outcome measurement – such as for substance use recurrence?

- Please use person first language throughout. Ex: page 10, ‘users’ and ‘misusers’ is not appropriate, please use terminology more consistent with ‘individuals who use…’

- For the screening outcome, how will a ‘screen’ be measured? By use of a validated tool within context of a patient-provider interview?

- For the ‘contacts’ part under fidelity procedures, is this referring to contacts between providers?

- For the SMART ECHO measures – such as the leveraging of technologu, quality checklist, etc. Have these measures been used by other ECHO studies in other areas of medicine? If yes, references would be great.

- For the sample size descriptions, I don’t see any discussion that these #s will be feasible at the sites? For example, 90 eligible participation per cluster for hypothesis one paragraph is not a small number – will that be feasible? And how do you know that?

- The description of the qualitative methods is lacking – sample size justification, has the interview guide been developed – and based on what, planned analysis, etc. etc.

- Lastly, and very importantly, after reading the whole article it is not clear to me if the purpose of this study is to show which of these 2 models are BETTER at accomplishing the intended outcomes. Or is it rather to show that both are feasible and attain the intended outcomes?

TABLES/FIGURES

Figure 1: Is the NIDA Quick Screen what needs to be done for the screening outcome? How did you choose this one? In other studies, this is not superior to other tools such as 4Ps and SURP when restricted to pregnancy. Also, what is ‘Assigned OUD treatment’? Please delineate in table what are patient and what are provider measures.

Figure 2: In the text it says patients can be enrolled up to 34 weeks, but there is a 26 week questionnaire. What will be done for those who enroll after 26 weeks? Also, add ‘patient’ to title.

Table 1: Would be great if how these outcomes are specifically defined is added to table, or at least in footnotes.

Table 2: Same here, at least for the measures that are not self-explanatory

Reviewer #2: Thank you for the opportunity to review this protocol. The work is important and addresses an area of clear need. My main concern with the paper is the lack of clarity around the comparison and hypothesis. The study is a direct comparison of two models of care using a cluster RCT. There is no comparison to either usual care of gold standard care. The authors appear to hypothesize that the case manager model will be more effective, but this is buried in the paper and is not entirely clear. I think this needs to be explicitly addressed and the hypotheses explicitly stated and restated where necessary.

I have only a small number of comments which I hope will bring better clarity for the reader.

1. SMART is already a commonly used acronym in addiction treatment. SMART-Recovery (Self Management And Recovery Training) is an internationally recognised self-help group and this project could be confused with that (for e.g., its impact on pregnant women). This is a minor point only, but the authors may wish to consider a way of distinguishing the two terms.

2. I found the first paragraph of the introduction difficult to follow and the third sentence “Approximately 7%...use of their medication” unnecessary and should be removed. The key issue to my mind is that, due to the opioid crisis in the US, there has been an increasing number of women with opioid use disorder who are pregnant, but there is paucity of specialist obstetric/addiction providers. The introduction could be more clearly and plainly laid out.

3. Second paragraph under Methods and Design – there is superfluous detail regarding Institutional Review Board approval, for e.g., I do not think the address of the IRB is needed.

4. Minor error paragraph 1, page 6, “buprenorphine-wavered” should be “buprenorphine-waivered”.

5. More detail is required for site recruitment. Presumably the obstetric centres recruited were approached directly by the study team? Can the authors provide a little more detail on how an “obstetric centre” is defined?

6. Individual participants (i.e., eligible women) will have to provided consent, what data if any will be collected on those who do not provide consent?

7. Page 14, it would be beneficial for the authors to restate each of the hypotheses to contextualise the power calculations.

8. How many trial participants are expected to be interviewed in the qualitative arm? A little more detail in this section would be useful.

9. Will there be any health economics data collected?

10. Will social outcomes be measured? For example, how many of the infants are removed by social services? Or what level of social service involvement for each group?

7. PLOS authors have the option to publish the peer review history of their article (what does this mean?). If published, this will include your full peer review and any attached files.

Reviewer #1: No

Reviewer #2: No

---

## [Author Response · Author response to Decision Letter 0]

21 Oct 2021

Response to Reviewers

We would like to thank the reviewers for their time and careful review of our manuscript. We have made the suggested changes as outlined below. 

Reviewer #1: This study presents a protocol for a trial aimed to investigate best practices for supporting OB care teams to integrate OUD treatment into OB care for pregnant women. I am happy to hear this trial is funded and going to take place, as we are desperately in need of evidence-based approaches at the ‘practice’ level to support providers in addressing the overdose crisis, especially for a priority population like pregnant and parenting people. Below I outline some queries and suggestions to strengthen the manuscript.

TEXT

- For the SMART ECHO program, is there a set curriculum that will be carried out across institutions? Will it be one ‘hub’ for all participating study practices? Or will there be multiple?

In the spirit of the original Project ECHO model, the SMART ECHO will have one hub for all study practices randomized to this model. We have clarified this in page 6 of the manuscript.

- For the CC-OP, the registry is brought up in the methods multiple times, coming off as critical to proper data collection as well as effectiveness of the intervention. Later on in the manuscript it starts to become more clear what this registry is – but would be helpful earlier on. For example, who is in charge of keeping this registry? 

We have added further clarification about the registry to page 7:

“The registry is a secure web-based patient tracking system that is updated twice a month by the case manager.”

- Will there be one case manager per site? Full time position? What qualifications will the CM have?

We have clarified this on page 8:

“The collaborating physicians identify a part-time mid-level clinician, such as a nurse, social worker or research assistant, who is trained to be the CM for CC-OP or data support person for SMART ECHO.”

- Under primary outcomes, it says ‘treatment for opioid misuse’ – to be consistent with methods, I believe this should be OUD

We agree, thank you. This has been corrected on page 9.

- Will the treatment retention outcome be censored at 3 months? How will you handle that the ‘end of the study’ is 3 months but people may start treatment at different times

The retention outcome will not be censored at 3 months. We are tracking gestational age at the time of study entry as well as time in the study. We did not plan to include these variables in our primary analyses, but we will examine if they are balanced between treatment groups and can run secondary analyses to see if they have an impact on treatment effect. 

- Also, it is imperative that a justification for why this study will stop measuring outcomes at 3 months postpartum, rather than 12 months, is recommended – given that 7-12 months is the highest risk time for postpartum women with OUD for overdose, as an example.

We limited our follow-up period to three months postpartum, because the goal of the study is to evaluate interventions that support obstetrical practices in caring for perinatal patients that have an OUD, and obstetrical providers usually do not closely follow patients beyond three months postpartum. We are, however, monitoring patients’ ability to connect with other treaters after delivery, and have included training around referrals and warm handoffs for both conditions. We do appreciate that the risk of overdose is higher in the late postpartum period (>6 months) and there is certainly a need to develop and investigate interventions that can address this. Unfortunately, this was beyond the scope of this study.

- How will you measure if a patient was ‘offered’ MOUD? By provider documentation – and then chart abstraction? If yes, how will you mitigate information bias?

We measure whether MOUD was offered via patient self-report at a baseline survey and monthly study calls with the research assistant (see page 14).

- Will substance use recurrence be measured only by urine drug test, or also by self-report? Post COVID, we are doing a lot more telehealth, so how will you take that into consideration for outcome measurement – such as for substance use recurrence?

We are relying on patient self-report for this secondary outcome. We are collecting the Timeline Follow-Back as described on page 13. 

- Please use person first language throughout. Ex: page 10, ‘users’ and ‘misusers’ is not appropriate, please use terminology more consistent with ‘individuals who use…’

Thank you for pointing this out, we have addressed this on page 10 and have ensured it is consistent throughout the manuscript.

- For the screening outcome, how will a ‘screen’ be measured? By use of a validated tool within context of a patient-provider interview?

Screening is a required element for all sites randomized to CC-OP via a validated screening tool. We have included the following on page 7:

“Universal screening is done via a validated tool (e.g,. the NIDA quick screen, the 4P’s Plus, the SURP-P, etc.), selected by the individual practices and integrated to the individual practice workflow.”

For the purposes of assessing screening as part of the CC-OP fidelity measures, we have included the following to page 11:

“Screening is considered completed if patients were asked about substance use using a validated scale and documented in the medical record.”

- For the ‘contacts’ part under fidelity procedures, is this referring to contacts between providers?

This refers to providers contacting patients. This has been clarified on page 11.

- For the SMART ECHO measures – such as the leveraging of technology, quality checklist, etc. Have these measures been used by other ECHO studies in other areas of medicine? If yes, references would be great.

We are collaborating with the Weitzman Institute (https://www.weitzmaninstitute.org/project-echo) to deliver SMART ECHO. They are an accredited replication center for Project ECHO. We have now included this on page 6.

We have also included the following reference on page 11: 

Khatri K, Haddad M, Anderson D. Project ECHO®: Replicating a Novel Model to Enhance Access to Hepatitis C Care in a Community Health Center. Journal of Health Care for the Poor and Underserved. 2012;24(2):850-858.

- For the sample size descriptions, I don’t see any discussion that these #s will be feasible at the sites? For example, 90 eligible participation per cluster for hypothesis one paragraph is not a small number – will that be feasible? And how do you know that?

This has now been added to page 14:

“The proposed sites are primarily in high-need, urban settings. Combined, the sites in our study have an average of 36,995 deliveries per year. Based on the rates of OUD in pregnant women in MA (13.1/1000) and CT (10/1000) this would mean an average of 430 pregnant women per year with an OUD would be served by these locations for a total of 1,075 eligible women (or 90 per site) over the 2.5 years of study recruitment.”

- The description of the qualitative methods is lacking – sample size justification, has the interview guide been developed – and based on what, planned analysis, etc. etc.

A separate manuscript is being written to describe the details of the qualitative procedures and protocol. We have added this to the manuscript on page17.

- Lastly, and very importantly, after reading the whole article it is not clear to me if the purpose of this study is to show which of these 2 models are BETTER at accomplishing the intended outcomes. Or is it rather to show that both are feasible and attain the intended outcomes?

In order to clarify this, we have included the following on page 5:

“This protocol was in response to a Patient-Centered Outcomes Research Institute (PCORI) funding initiative addressing the following question: “What is the comparative effectiveness of different strategies for providing support or coordination of services for components of medication-assisted treatment (MAT) (induction and/or psychosocial services) to providers who offer office-based opioid treatment (OBOT) to pregnant women, in terms of maternal and neonatal outcomes?” (https://www.pcori.org/funding-opportunities/announcement/medication-assisted-treatment-cycle-2-2018).

TABLES/FIGURES

Figure 1: Is the NIDA Quick Screen what needs to be done for the screening outcome? How did you choose this one? In other studies, this is not superior to other tools such as 4Ps and SURP when restricted to pregnancy. Also, what is ‘Assigned OUD treatment’? Please delineate in table what are the patient and what are the provider measures. 

We have corrected the figure as the NIDA Quick Screen is not the required screening tool. As noted earlier, practices determine the validated tool they utilize for screening. The provider measures are bolded and have a footnote to identify them as such.

We have provided further clarification in the figure legend:

1. Participant will enter the study at different point of pregnancy. Therefore, it is likely that participants will complete varying numbers of visits between screening and the 26-week visit, and the 36-week visit and birth. This is expected and is not considered a protocol deviation. Visit 1 will follow immediately after the consenting process.

2. All participating centers will administer a validated substance use screening tool, such as the NIDA Quick Screen, 4Ps Plus or equivalent validated instrument as standard of care. The tool utilized is determined by the clinical site. 

3. The DSM-5 OUD Checklist will be administered to any patients who have a positive substance use screen as part of the standard of care. 

4. The Timeline Followback (TLFB) will be conducted monthly during the length of the study. The number of TLFB assessments will vary depending on when the patient enters the study and when they give birth. 

5. Birth outcomes include: low birth weight, resuscitation at delivery, fetal demise, preterm birth, duration of hospitalization. 

6. Provider measures: The Substance Abuse Attitude Scale, the Physicians Worklife Survey and the Qualitative Interviews will be completed by providers and done prior to the first participant being enrolled at the site and after the last participant is enrolled at the site. 

Figure 2: In the text it says patients can be enrolled up to 34 weeks, but there is a 26 week questionnaire. What will be done for those who enroll after 26 weeks? Also, add ‘patient’ to title.

The questions from the 26-week visit are incorporated to the baseline assessment. We have added this to the figure:

“If a patient-participant is enrolled after 26 weeks’ gestation and before 36 weeks’ gestation, the items from the 26-week visit are added to the baseline assessment.”

Table 1: Would be great if how these outcomes are specifically defined is added to table, or at least in footnotes.

This was added.

Table 2: Same here, at least for the measures that are not self-explanatory

This was added.

Reviewer #2: Thank you for the opportunity to review this protocol. The work is important and addresses an area of clear need. My main concern with the paper is the lack of clarity around the comparison and hypothesis. The study is a direct comparison of two models of care using a cluster RCT. There is no comparison to either usual care of gold standard care. The authors appear to hypothesize that the case manager model will be more effective, but this is buried in the paper and is not entirely clear. I think this needs to be explicitly addressed and the hypotheses explicitly stated and restated where necessary.

We appreciate the reviewers comment around the lack of clarity. This was also noted by reviewer 1 above, and we have addressed this with the additional information provided on page 5. 

I have only a small number of comments which I hope will bring better clarity for the reader.

1. SMART is already a commonly used acronym in addiction treatment. SMART-Recovery (Self Management And Recovery Training) is an internationally recognised self-help group and this project could be confused with that (for e.g., its impact on pregnant women). This is a minor point only, but the authors may wish to consider a way of distinguishing the two terms.

For clarity in the manuscript, we have referred to our trial “Project SMART”

2. I found the first paragraph of the introduction difficult to follow and the third sentence “Approximately 7%...use of their medication” unnecessary and should be removed. The key issue to my mind is that, due to the opioid crisis in the US, there has been an increasing number of women with opioid use disorder who are pregnant, but there is paucity of specialist obstetric/addiction providers. The introduction could be more clearly and plainly laid out.

The opening paragraph has been re-written:

“Between 1999 and 2014 the point prevalence of an opioid use disorder (OUD) in pregnant women increased from 1.5 to 6.5 per 1000 deliveries, which is in line with rates of opioid prescribing in the general population.(Haight, Ko, Tong, Bohm, & Callaghan, 2018) Thus, it is not surprising that between 2007 and 2016, pregnancy-associated mortality resulting from overdose more than doubled in the US, from 1.3 to 4.2 deaths per 100,000 live births. In many states, overdose is the leading cause of maternal morbidity and mortality.(Cleveland et al., 2020) Unfortunately, outpatient providers of medication treatment for OUD (MOUD) are less likely to treat pregnant women compared to non-pregnant women (75% vs 91%).(Patrick et al., 2018)” 

3. Second paragraph under Methods and Design – there is superfluous detail regarding Institutional Review Board approval, for e.g., I do not think the address of the IRB is needed.

This has been removed.

4. Minor error paragraph 1, page 6, “buprenorphine-wavered” should be “buprenorphine-waivered”.

This has been corrected.

5. More detail is required for site recruitment. Presumably the obstetric centres recruited were approached directly by the study team? Can the authors provide a little more detail on how an “obstetric centre” is defined?

The following has been added to page 8:

“Obstetrical clinics of any type, private practices, hospital-based clinics, or Federally Qualified Health Centers, were approached by the study principal investigators.”

6. Individual participants (i.e., eligible women) will have to provided consent, what data if any will be collected on those who do not provide consent?

This has been added to page 13:

“If consent is not provided, only basic demographic information, number of weeks pregnant and the reason for disinterest, if known, is collected.”

7. Page 14, it would be beneficial for the authors to restate each of the hypotheses to contextualise the power calculations.

This has been addressed on pages 14-15.

8. How many trial participants are expected to be interviewed in the qualitative arm? A little more detail in this section would be useful.

We aim to interview 1-2 patient-participants per clinical site. This has been added to page 17. 

9. Will there be any health economics data collected?

No health economic data will be collected.

10. Will social outcomes be measured? For example, how many of the infants are removed by social services? Or what level of social service involvement for each group?

 Yes, this is collected as part of the 3-month follow-up interview, this has been added to page 14.

---

## [Decision Letter · Decision Letter 1]

10 Dec 2021

Support Models for Addiction Related Treatment (SMART) for Pregnant Women: Study Protocol of a Cluster Randomized Trial of Two Treatment Models for Opioid Use Disorder in Prenatal Clinics

PONE-D-21-07384R1

Dear Dr. Forray,

We’re pleased to inform you that your manuscript has been judged scientifically suitable for publication and will be formally accepted for publication once it meets all outstanding technical requirements.

Kind regards,

Yukiko Washio, Ph.D.

Academic Editor

PLOS ONE

Additional Editor Comments (optional):

Reviewers' comments:

Reviewer's Responses to Questions

**Comments to the Author**

1. Does the manuscript provide a valid rationale for the proposed study, with clearly identified and justified research questions?

Reviewer #1: Yes

Reviewer #2: Yes

2. Is the protocol technically sound and planned in a manner that will lead to a meaningful outcome and allow testing the stated hypotheses?

Reviewer #1: Yes

Reviewer #2: Yes

3. Is the methodology feasible and described in sufficient detail to allow the work to be replicable?

Reviewer #1: Yes

Reviewer #2: Yes

4. Have the authors described where all data underlying the findings will be made available when the study is complete?

Reviewer #1: Yes

Reviewer #2: No

5. Is the manuscript presented in an intelligible fashion and written in standard English?

Reviewer #1: Yes

Reviewer #2: Yes

6. Review Comments to the Author

You may also provide optional suggestions and comments to authors that they might find helpful in planning their study.

Reviewer #1: Thank you authors for addressing the reviewer comments. The paper is a lot more clear, understandable, and thus will be an even better addition to the literature for a general audience. A couple minor things below to consider before publication:

- As a clinician, it is still not clear to me the role that the case manager will play in the trajectory between screening > assessment by provider for OUD > treatment options presented > treatment initiation. Will the CM be the one to contact the pt if the screen is positive? Or does the provider refer the patient to the CM after discussing a positive screen with the patient? Then the CM coordinates follow up apts for the OUD assessment? etc. ? Thank you for clarifying.

- How long after a patient 'receives' the diagnosis of OUD will the pt be recruited for study participation? It can be challenging, and overwhelming, for patients entering care let alone OUD treatment for the firs time during pregnancy - and then also being approached for a research study can further that. Is there a limit' on how long after OUD dx people remain eligible to participate - the more flexibility the better, likely, to allow for optimal recruitment rates.

- Thank you for adding the information about the specifics of the hypotheses. It is interesting (yet not surprising) that all the hypotheses lean towards the case management arm outperforming the ECHO arm. Yet based on the descriptions of the trial interventions/arms earlier in the paper, I was coming into this section of the manuscript thinking they were likely going to be on par with one another. Potentially adding one sentence to the discussion or elsewhere stating WHY the authors hypothesize that the CM arm will achieve better patient outcomes than the ECHO arm will be useful for context.

-A chunk of the discussion revolves around payment/costs, which is very important. What is in the protocol, data collection wise, to ensure that a proper, robust cost benefit analysis will be able to be performed at the conclusion of this trial?

Reviewer #2: The authors have generally and adequately addressed my queries.

However, consistent with my earlier comment, given the potential confusion with SMART recovery, I suggest the authors reconsider the title by either removing the acronym SMART or add in the word “Project” as a prefix to the acronym. I believe this would benefit the readership and avoid confusion, but the suggestion is at the author's (or the editor) discretion.

Likewise, I appreciate that a health economics component is not part of the trial and I think that is disappointing as any intervention for this group will need to be economically viable for the healthcare system and is therefore an important piece of the effectiveness puzzle. As such, I would encourage the authors to at least consider this point in the discussion. Again, however, how much the authors engage with this should be a the author’s discretion.

7. PLOS authors have the option to publish the peer review history of their article (what does this mean?). If published, this will include your full peer review and any attached files.

Reviewer #1: No

Reviewer #2: No

---

## [Editor Report · Acceptance letter]

5 Jan 2022

PONE-D-21-07384R1 

Support Models for Addiction Related Treatment (SMART) for Pregnant Women: Study Protocol of a Cluster Randomized Trial of Two Treatment Models for Opioid Use Disorder in Prenatal Clinics 

Dear Dr. Forray:

I'm pleased to inform you that your manuscript has been deemed suitable for publication in PLOS ONE. Congratulations! Your manuscript is now with our production department. 

Kind regards, 

on behalf of

Dr. Yukiko Washio 

Academic Editor

PLOS ONE